# Optimizing Therapies in Heart Failure: The Role of Potassium Binders

**DOI:** 10.3390/biomedicines10071721

**Published:** 2022-07-16

**Authors:** Pietro Scicchitano, Massimo Iacoviello, Francesco Massari, Micaela De Palo, Pasquale Caldarola, Antonia Mannarini, Andrea Passantino, Marco Matteo Ciccone, Michele Magnesa

**Affiliations:** 1Cardiology Section, Hospital “F. Perinei” Altamura (BA), 70022 Altamura, Italy; franco_massari@libero.it; 2Cardiology Unit, Department of Medical and Surgical Sciences, University of Foggia, 71122 Foggia, Italy; massimo.iacoviello@unifg.it (M.I.); michele.magnesa17@gmail.com (M.M.); 3Cardiac Surgery Unit, Azienda Ospedaliero-Universitaria Policlinico Bari, 70124 Bari, Italy; micaela.depalo85@gmail.com; 4Cardiology Section, Hospital “S. Paolo” Bari, 70123 Bari, Italy; pascald1506@gmail.com; 5Division of University Cardiology, Cardiothoracic Department, Policlinic University Hospital, 70124 Bari, Italy; antonia.mannarini@gmail.com; 6Division of Cardiology and Cardiac Rehabilitation, Scientific Clinical Institutes Maugeri, IRCCS Institute of Bari, 70124 Bari, Italy; andrea.passantino@icsmaugeri.it; 7Section of Cardiovascular Diseases, Department of Emergency and Organ Transplantation, University of Bari, 70124 Bari, Italy; marcomatteo.ciccone@uniba.it

**Keywords:** patiromer, sodium zirconium cyclosilicate, heart failure, prognosis, management

## Abstract

Heart failure (HF) is a worrisome cardiac pandemic with a negative prognostic impact on the overall survival of individuals. International guidelines recommend up-titration of standardized therapies in order to reduce symptoms, hospitalization rates, and cardiac death. Hyperkalemia (HK) has been identified in 3–18% of HF patients from randomized controlled trials and over 25% of HF patients in the “real world” setting. Pharmacological treatments and/or cardio-renal syndrome, as well as chronic kidney disease may be responsible for HK in HF patients. These conditions can prevent the upgrade of pharmacological treatments, thus, negatively impacting on the overall prognosis of patients. Potassium binders may be the best option in patients with HK in order to reduce serum concentrations of K^+^ and to promote correct upgrades of therapies. In addition to the well-established use of sodium polystyrene sulfonate (SPS), two novel drugs have been recently introduced: sodium zirconium cyclosilicate (SZC) and patiromer. SZC and patiromer are gaining a central role for the treatment of chronic HK. SZC has been shown to reduce K^+^ levels within 48 h, with guaranteed maintenance of normokalemia for up to12 months. Patiromer has resulted in a statistically significant decrease in serum potassium for up to 52 weeks. Therefore, long-term results seemed to positively promote the implementation of these compounds in clinical practice due to their low rate side effects. The aim of this narrative review is to delineate the impact of new potassium binders in the treatment of patients with HF by providing a critical reappraisal for daily application of novel therapies for hyperkalemia in the HF setting.

## 1. Introduction

Globally, heart failure (HF) affects approximately 23 million people worldwide including more than 15 million people in Europe [1]. Statistics have revealed a prevalence rate of 1–3% in the general adult population and an incidence equal to 1–20 cases per 1000 person-years, and mortality ranging from 2 to 3% at 30 days from hospital discharge and from 50 to 75% at five-year follow-up [2].

International guidelines [3] recommend the need to modulate the main determinants of HF progression, namely the renin–angiotensin–aldosterone system (RAAS), the autonomic system, and the natriuretic peptide system, by optimizing therapies to the maximum tolerated dose [3].

Several trials [4,5,6,7,8,9,10,11,12] have demonstrated the benefits related to RAAS inhibitors (RAASi) in counteracting the negative evolution of HF, as they have demonstrated substantial reductions in hospitalization for HF, cardiovascular mortality, and improvement in the New York Heart Association (NYHA) classification system. A network meta-analysis [13,14] substantially proved the need for an early combination of different classes of drugs that could inhibit, on parallel, the different actors of HF progression and improve the risk for all-cause mortality by at least 70%. Therefore, optimizing pharmacological treatments in HF by up-titrating doses of all HF drugs (i.e., angiotensin converting enzyme inhibitors (ACEis), angiotensin-II receptor blockers (ARBs), mineralcorticoid receptor antagonists (MRAs), angiotensin receptor-neprilysin inhibitors (ARNIs), beta-blockers, and sodium-glucose cotransporter-2 inhibitors (SGLT2is)) has the main goal to be achieved by clinicians [3].

Beyond the direct and indirect pharmacological effects of these drugs on the heart, kidneys are further, secondary targets to be affected. Compounds that can interact with RAAS or SGLT2 may alter kidney function. As kidneys are the main regulators for ionic balance, one could suppose the possible impact of such drugs on the homeostasis (secretion and excretion) of ions maintained by the kidneys. Potassium is mostly involved in such a deregulation and the kidneys are responsible for the excretion of 90% of potassium, thus, conditions that have an impact on the complex renal mechanisms involved in potassium homeostasis account for serum increases of this ion [15,16].

The aims of this narrative review include: (1) To provide an overview of the prevalence rate and impact of hyperkalemia (HK) in the prognosis of HF, (2) to elucidate current therapies for counteracting hyperkalemia in HF, and (3) to provide a critical reappraisal for daily application of novel therapies for hyperkalemia in the HF setting.

Finally, in this narrative review, we provide an overview of advances in treating HK in patients with HF and propose a dedicated clinical flow chart for physicians to use for guiding the decision-making process in the management of HF patients.

## 2. Search Strategies

We performed the literature search by consulting the main scientific databases: MEDLINE, MEDLINE In-Process and Other Non-Indexed Citations, EMBASE, PubMed, and the Cochrane Central Register of Controlled Trials through the Ovid interface.

We adopted the following terms for the research: “hyperkalemia”, “serum potassium binders”, “heart failure”, “prognosis”, “management”, “sodium polystyrene sulfonate”, “sodium zirconium cyclosilicate”, “patiromer sorbitex calcium”, “incidence”, and “adverse events”. Different combinations of all the terms were used in order to better perform the final collection of the papers to be included in this study.

Two researchers (P.S. and M.M.) compared the results of the research and eliminated duplicate studies. All types of studies, i.e., randomized controlled trials, observational studies, retrospective studies, etc., that dealt with the use of potassium binders within the HF setting were analyzed and the results were included in the final manuscript.

## 3. Hyperkalemia in HF: Definition, Prevalence, and Prognosis

Hyperkalemia is defined when serum potassium (K^+^) levels exceed 5 mEq/L [17]. We can classify HK into: “mild” when serum K^+^ levels range between 5 and 5.5 mEq/L, “moderate” when serum K^+^ levels range between 5.5 and 6 mEq/L, and “severe” when serum K^+^ levels are higher than 6 mEq/L [17].

The prevalence of HK in the setting of HF is well-established [18]. The evaluation of results from randomized controlled trials (RCTs) dealing with HF in all its forms revealed an overall prevalence in any type of HK ranging from 3 to 18% [18]. The identification of this higher number of cases of HK in RCTs accounted for discontinuing HF therapies in 0.6–3.5% of cases [18].

Current real-world percentages are not dissimilar to those from RCTs. Data from 2,270,635 U.S. patients (2010–2014) highlighted a 1.57% prevalence rate of HK in the overall population and a 6.35% prevalence rate in those with chronic kidney disease (CKD) and/or heart failure, while the prevalence rate of CKD and/or HF was 48.43% in patients with HK [19]. Similar results have been reported in the Medicare population, with a 2.6–2.7% prevalence rate of HK in the overall population and a 8.9–9.3% prevalence rate of HK among patients with CKD and/or HF [20]. The data from Europe are not dissimilar to those from the USA, for example, in Italy, the prevalence rate of HK has been calculated to range from 6 to 10% in patients with HF [18]. The data from a recent Danish population-based cohort study revealed the prevalence rate of HK, in patients with congestive HF but normal kidney function (defined as estimated glomerular filtration rate (eGFR) to be >60 mL/min/1.73 m^2^), to br about 35% (mild HK), 13% (moderate HK), and 8% (severe HK) in relation to HK degree [21]. Indeed, in patients with congestive HF and severe CKD (eGFR 15–29 mL/min/1.73 m^2^), the prevalence of HK ranged from about 70% (mild HK) to 25% (severe HK) [21]. According to the SwedeHF (Swedish Heart Failure) Registry, about a quarter of patients with any type of HF (with reduced ejection fraction (HFrEF), mildly reduced ejection fraction (HFmrEF), and preserved ejection fraction (HFpEF)) suffer HK, thus, negatively impacting on the overall survival rate of these patients [22].

Higher plasma K^+^ levels could effectively impact on the prognosis of patients, and those suffering from HF in particular. The relationship between serum K^+^ concentration and the prognosis of patients with HF is similar to a U wave: lower and higher values than normal range negatively impact on the health of patients [23]. Indeed, the higher the plasma K^+^ concentration, the higher the incidence of all-cause mortality, cardiovascular death, death due to HF, and sudden cardiac death even after adjusting for confounding factors [23]. Although the data from the SwedeHF Registry have revealed mortality rates higher than 20% [22,23] in patients with HK and HF, the return to normokalemia might definitely improve the cumulative probability of all-cause mortality [23].

Many conditions can be considered to be independent predictors of high potassium levels, i.e., advancing age, history of diabetes, and decrease in eGFR. Indeed, drugs, such as ACEi/ARB or spironolactone, might be responsible for a 51% and 47% increase in serum K^+^ levels > 5.5 mEq/L [24]. A subanalysis from the ESC-EORP-HFA Heart Failure Long-Term Registry revealed that HK could prevent persistence on treatment or avoidance of target of the recommended dose of RAASi in patients with HFrEF, independently from the baseline value of K^+^ [25].

The main consequences of such data are related to the net increase in 1-year mortality and re-admission for HF: HF patients who discontinued or did not start RAASi showed a cumulative incidence in mortality at 1-year follow-up higher than 41%, while the rate of re-admission for HF decompensation at 1-year could exceed 64% [26]. Lisi et al. [27] observed a 75% mortality rate at 65-month follow-up when MRA was discontinued in patients with HFrEF.

## 4. Treatment of Hyperkalemia: Advances in Therapies

Physicians who are involved in the management of patients with HF should balance the need for prescribing RAASi at the maximum tolerated dose and the corresponding risk for hyperkalaemia, leading to the need for withdrawal or reduction of drug dosages which may negatively impact patients’ outcomes [28].

Treating HK is challenging due to limited pharmacological compounds that can be adopted within the outpatient setting. Nevertheless, recent advances in cardiovascular pharmacology have provided new insights into the general management of HK in HF. Beyond the “ancient” treatment using sodium polystyrene sulfonate, two drugs have been introduced for HK treatment: patiromer sorbitex calcium and sodium zirconium cyclosilicate.

The physiopathological background for the use of these compounds is related to an increase in intestinal excretion of K^+^. About 10% of ingested K^+^ is secreted by the gut; although most ingested K^+^ is absorbed in the small intestine (jejunum > ileum), the secretion is mediated by the colon [29,30]. In the case of a decrease in the excretion of the K^+^ from the kidney due to renal failure/use of pharmaceutical compounds, K^+^ excretion from the intestine can increase to 50% [29,30]. Based on these principles, potassium binders act at the intestinal level for balancing plasma K^+^ concentrations (Table 1).

### 4.1. Sodium Polystyrene Sulfonate

Sodium polystyrene sulfonate (SPS) is a cation-exchange resin; its chemical structure is formed by benzene, diethenyl-polymer, with ethenylbenzene-sulfonated sodium salt [31,32]. The Food and Drug Administration (FDA) approved this compound in 1958. It can be orally or rectal taken, and the compound mainly acts at the level of the colon where it exchanges sodium for potassium. Indeed, SPS is a non-specific cation-exchange: the sodium ions in the compound may be displaced by calcium (Ca^2+^) and/or magnesium (Mg^2+^), thus, accounting for the lack of specificity of the drug [31,32]. This also accounts for the variability in the drug’s action onset, while 33% is the exchange skill of SPS bolus [31,32].

The literature contains scant data on the effectiveness and safety of SPS employment in clinical practice (Table 2).

Batterink et al. [33] evaluated the impact of SPS on patients with HK (K^+^ level between 5.0 and 5.9 mEq/L) as compared with a placebo. SPS reduced serum K^+^ levels by 0.14 mEq/L, but no significantly clinical effect was observed [33].When administered for 7 days in patients with CKD and mild hyperkalemia (5.0–5.9 mEq/L) at a dose of 30 g orally o.d., SPS was effectively able to significantly reduce serum K^+^ levels as compared with a placebo, with no significant increase in side effects [34]. Indeed, a dose of 60 g o.d. might be more effective than the 30/15 g o.d. or the use of a rectal dose of 30 g o.d. [35,36]. Specifically, it seems that the higher the serum K^+^ levels, the higher the absolute reduction in the plasma levels of the ions within 24 h after the administration, although conflicting results are in the literature about this datum [37]. Indeed, data from small groups of patients who suffered CKD, used RAASi, and showed at least one episode of HK, low dose (15 mg o.d.) SPS might effectively reduce serum K^+^ levels when administered for long-term follow-up [38,39].

Nevertheless, SPS has demonstrated adverse serious side effects [40]. Case reports have reported gastrointestinal injuries such as colitis and necrosis; however, a recent analysis in the literature by Holleck et al. [40] did not find significantly higher rates in intestinal necrosis, although the composite outcome of severe gastrointestinal adverse events was significantly increased. The FDA has provided a warning about the use of sorbitol in concomitance with SPS as it can increase the risk for gastrointestinal necrotic events [31,32].

**Table 2 biomedicines-10-01721-t002:** Main characteristics of studies on sodium polystyrene sulfonate.

Study	N. of pts	Type of pts	Design	Approach	Follow-Up	Results
Batterink et al., 2015 [33]	138	Serum K^+^ between 5.0 and 5.9 mEq/L	Retrospective observational study	72 control group 66 treatment group (dose 15 or 30 g)	24 h	ΔK^+^ 6 h: −0.44 ± 0.29 mEq/L; ΔK^+^ 24 h: −0.58 ± 0.39 mEq/L (*p* = 0.026) No difference between patients 15 and 30 g of SPS
Lepage et al., 2015 [34]	33	Outpatients with CKD and serum K^+^ between 5.0 and 5.9 mEq/L	RCT	Placebo or SPS 30 g orally o.d. for 7 days	7 days	SPS ↓ K^+^ levels (mean difference between groups: −1.04 mEq/L; 95% CI, −1.37 to −0.71). Normokalemia in 73% of pts with SPS Trend toward ↑ electrolytic disturbances and GI side effects in SPS group.
Mistry et al., 2016 [35]	118	Patients who received SPS	Retrospective observational study	SPS 15, 30, and 60 g oral and 30 g Rectal	12 h	↓ K^+^ by 0.39, 0.69, 0.91, and 0.22 mEq/L following 15, 30, and 60 g oral doses and a 30 g rectal dose of SPS, respectively. 50% vs. 23% remained hyperkalemic in the 15 g group vs. 60 g group (*p* = 0.018) All patients in the rectal group remained hyperkalemic. No patient with postdose hypokalemia.
Sandal et al., 2012 [37]	135	Patients who received SPS	Retrospective observational study	15 and 30 g o.d.	24 h	↓ K^+^: 16.7% (*p* < 0.001) within 24 h. No change in serum creatinine Patients with higher baseline K^+^ (≥5.6 mEq/L) better reduced K^+^ (>4%) than those with baseline K^+^ < 5.6 mEq/L (*p* = 0.32). No significant difference between 15 g and 30 g. 13 deaths, of which one due to ischemic colitis
Kessler et al., 2011 [36]	122	Patients with K^+^ >5.1 mEq/L.	Retrospective observational study	15, 30, 45, and 60 g o.d.	N/A	↓ K^+^: 0.82 ± 0.48 mEq/L in 15 g group 0.95 ± 0.47 mEq/L in 30 g group 1.11 ± 0.58 mEq/L in 45 g group 1.40 ± 0.42 mEq/L in 60 g group. Normal range K^+^ in 94% patients
Chernin et al., 2012 [38]	14	CKD and heart disease on RAAS-I treatment after at least 1 episode of K^+^ ≥ 6.0 mEq/L	Prospective, longitudinal study	15 g o.d.	14.5 months	None developed colonic necrosis or life-threatening events attributed to SPS use. Mild hypokalemia in 2 patients No further episodes of hyperkalemia were recorded
Georgianos et al., 2017 [39]	26	Outpatients with stages 3–4 CKD	Retrospective observational study	15 g o.d.	15.4 months	↓ K^+^: from 5.9 ± 0.4 to 4.8 ± 0.5 mEq/L (*p* < 0.001) Slight ↑ Na^+^: 139.5 ± 2.9 vs. 141.2 ± 2.4 mEq/L (*p* = 0.006). ~Ca^2+^ and phosphate No episode of colonic necrosis or other serious adverse events 1 patient had gastrointestinal intolerance.

Abbreviations: Ca^2+^, serum concentration calcium; CKD, chronic kidney disease; g, grams; CI, confidence interval; GI, gastrointestinal; h, hours; K^+^, serum concentration potassium; Na^+^, serum concentration sodium; o.d., once daily; RCT, randomized controlled trials; SPS, sodium polystyrene sulfonate. ↑: increase; ↓: decrease.

### 4.2. Sodium Zirconium Cyclosilicate (SZC)

SZC is an inorganic cation exchanger which is composed of a uniform crystalline structure with micropores that are responsible for the entrapment of the cations [41]. Specifically, the chemical structure of SZC can preferentially address an exchange between hydrogen and sodium monovalent cations such as potassium or ammonium [41]. The engagement with Ca^2+^ and Mg^2+^ is less favourable: this might be related to the ionic dimension of the K^+^ and ammonium and the complex architecture of the crystalline structure [41]. Experimental studies have demonstrated that SZC interacts with K^+^ 25-fold more selectively than Ca^2+^ or Mg^2+^ (SPS has a 0.2–0.3-fold selectivity for both the di-cations) [41]. Laboratory data have revealed that SZC has a 9.3-fold higher affinity for K^+^ than SPS and a 125-fold more selectivity for K^+^ than SPS [41].

The compound is not orally absorbed and it is completely eliminated via feces, thus, there is no systemic dissemination for SZC. Its specific action is performed in the gastrointestinal tract as a whole; therefore, the drug reduces the absorption of K^+^ and eliminates it via feces [41].

Packham et al. [42] considered patients with HK, as defined by serum K^+^ levels between 5.0 and 6.5 mEq/L, to identify a correct dose for SZC in this setting (Table 3).

Patients were randomized to receive 1.25 g, 2.5 g, 5 g, or 10 g of ZS-9 or a placebo t.i.d. during the initial 48 h (initial phase); then, those who reached serum K^+^ levels between 3.5 and 4.9 mEq/L were randomized 1:1 to the placebo or the corresponding SZC dose o.d. (maintenance phase). All of the dosages succeeded in reducing serum K^+^ levels to normal values within 48 h, but only 5 g and 10 g doses maintained K^+^ levels in the normal range. Interestingly, the 10 g dose was able to significantly reduce K^+^ levels after 1 h from administration [42]. In the HARMONIZE trial [43], patients with serum K^+^ levels > 5.1 mEq/L were randomized to 10 g t.i.d. within 48 h, then to 5, 10, and 15 g o.d. for the following 28 days: 10 g t.i.d. reduced serum K^+^ levels to a normal range in 84% of patients within 24 h and in 98% of patients within 48 h, while all the regimes maintained K^+^ levels in a normal range during the 28 days after. Similar results were obtained from the HARMONIZE Global trial [44]. Indeed, when extending the administration of SZC to 337 days, as in the HARMONIZE extension trial [45], 84.3% of patients maintained K^+^ levels < 5.1 mEq/L and 98% of patients maintained serum K^+^ levels < 5.5 mEq/L. Such data were independent from kidney function. Roger et al. [46] demonstrated that 82% and 90% of patients with baseline eGFR < 30 and ≥30 mL/min/1.73 m^2^ succeeded in maintaining normokalemia after 10 g three times daily for 24–72 h, followed by once daily SZC 5 g for ≤12 months. SZC was also able to return K^+^ levels to normokalemia even in end-stage renal disease (ESRD) patients who were on hemodalysis. More than 40% of patients were able to return to a normal range of K^+^ with a 5 g administration which could be increased to 15 g in the DIALIZE trial [47].

Indeed, there are little data are on the efficacy of SZC in patients with HF who cannot optimize their pharmacological regimen due to HK conditions. A subanalysis from the HARMONIZE trial involved patients with HF history, with or without RAASi therapy, who had to continue their pharmacological treatments [48]. The results pointed out that 83%, 89%, and 92% of patients with HF and in therapy with 5 g, 10 g, and 15 g SZC, respectively, reached normal K^+^ levels during the maintenance phase [48]. Nevertheless, the small sample size (*n* = 94), the lack of differentiation of HF subtype, as well as the reduced number of patients with optimized therapy (only 69% of patients were on RAASi) reduced the evaluation of the impact of SZC in patients with HF. Imamura et al. [49] retrospectively evaluated 24 patients with left ventricle ejection fraction (LVEF) < 50% who were treated with SZC for 3 months (5 g, 10 g, and 15 g o.d.). They observed reductions in plasma levels of K^+^, improvement in the pharmacological treatments with RAASi, and amelioration in LVEF after 3 months [49]. Interesting insights are expected from the results of the ongoing Lokelma for RAAS Maximisation in CKD & Heart Failure (LIFT) trial [50]. The LIFT trial will include patients with LVEF < 40%, NYHA class II–IV, with serum K^+^ levels of 5.0–5.5 mEq/L, CKD (as defined by eGFR < 60 mL/min/1.73 m^2^), with none or a submaximal dose of ACEi/ARB and/or MRA [50]. Patients will undergo treatments with SZC 10 g t.i.d. for 48 h, then 5 or 10 g o.d. in agreement with serum K^+^ at each visit; meanwhile, RAASi therapy will be upgraded to the maximum tolerated dose. This trial is in the recruiting phase [51]. Further studies are needed in order to better evaluate the impact of SZC in HF patients with HK and the impact on the clinical outcomes of these individuals.

### 4.3. Patiromer

Patiromer was approved by the FDA in 2015 [52] and by the European Medicines Agency (EMA) in 2017 [53]. The complex chemical structure of the compound is formed by a carboxylic acid compound substituted at the alpha position with fluorine, comprising a copolymer of 2-fluoroacrylic acid, divinylbenzene, and 1,7-octadiene [54]. Sorbitol is added to patiromer but, differently from SPS, its concentration is 5- to 10-fold lower than SPS [54]. It has been calculated that patiromer has 1.5- to 2.5-fold higher skill in binding K^+^ than other polymers [54]. Calcium represents the cation for the exchange with K^+^ and this exchange is mostly performed in the large intestine. There is no absorption of this compound, thus, it can not pass into the systemic fluids.

Patiromer has been included in clinical studies in order to evaluate its impact on HK and, to some extent, in the HF setting (Table 4).

The OPAL HK study [55] evaluated the impact of patiromer (initial dose 4.2 g or 8.4 g twice a day for 4 weeks, then those with normokalemia were randomized to patiromer or placebo for 8 weeks) in patients with CKD, receiving RAASi, and with serum K^+^ levels between 5.1 and 6.5 mEq/L. Seventy-six percent of patients showed normal K^+^ levels after 4 weeks of treatment; during the maintenance phase, only 15% of patients showed HK (vs. 60% of the placebo group) [55]. The effects of patiromer on serum K^+^ levels have been shown to remain when the compound was continued to be administered for a long period of time. The AMETHYST-DN trial just demonstrated lower K^+^ levelswhen patiromer was administered for 52 weeks [56].

The need for implementing therapies in those with HF has been provisionally considered in the PEARL HF trial [57]. Although the trial was not set for different types of HF, the aim was to implement therapies in those with chronic HF and HK, who discontinued RAASi or had CKD (eGFR < 60 mL/min/1.72 m^2^). Patients also underwent implementation of MRA therapy; 91% of patients reached the goal target of spironolactone with K^+^ levels maintained in a normal range [57]. A subanalysis of the AMETHYST-DN trial pointed out that the use of patiromer was safe and well-tolerated in patients with mild HF and on RAASi therapy [58]. Indeed, the DIAMOND trial will shed light on the efficacy of patiromer in patients with HF and LVEF ≤ 40% who will be started or continued on MRA titrated to 50 mg/day and other RAASi therapy to ≥50% target dose [59]. Meanwhile, a recent pooled analysis on 653 patients (214 with and 439 without HF) on patiromer treatment, coming from RCT populations, has demonstrated a consistent decrease in serum K^+^ level during therapy, with mild-to-moderate adverse events in one third of patients [60]. A retrospective analysis did not observe a significant increase in hospitalization rate for heart failure between patiromer and SZC users [61]. Furthermore, retrospective studies [62,63] have demonstrated contrasting results in the overall performance of patiromer, although more structured trials should be designed in order to address gaps in evidences.

**Table 4 biomedicines-10-01721-t004:** Main characteristics of studies on patiromer.

Study	N. of pts	Type of pts	Design	Approach	Follow-Up	Results
Weir et al., 2015 [55] OPAL HK trial	237	CKD, on RAASi, serum K^+^ 5.1–6.5 mEq/L	RCT	Patiromer (initial dose 4.2 g or 8.4 g b.i.d.) for 4 weeks (initial treatment phase). Those who reached serum K^+^ 3.8–5.1 mEq/L randomized to continue patiromer or switch to placebo.	12 weeks	At week 4: 76% pts reached serum K^+^ 3.8–5.1 mEq/L. Recurrence of hyperkalemia in the next 8 weeks (serum K^+^ ≥ 5.5 mEq/L) occurred in 15% pts on patiromer.Adverse effects: mild-to-moderate constipation (11% pts); hypokalemia (3% pts).
Bakris et al., 2015 [56] AMETHYST-DN trial	306	Pts type 2 DM, eGFR 15 to <60 mL/min/1.73 m^2^, serum K^+^ level > 5.0 mEq/L, RAASi	RCT	Stratified by baseline serum K^+^ into mild or moderate HK groups and received 1 of 3 randomized starting doses of patiromer (4.2 g b.i.d., 8.4 b.i.d., or 12.6 g b.i.d. (mild HK) or 8.4 g b.i.d., 12.6 g b.i.d., or 16.8 g b.i.d. (moderate HK)).	52 weeks	Mild group, reduction in K^+^: - 4.2 g b.i.d.: −0.35 mEq/L - 8.4 g b.i.d.: −0.51 mEq/L - 12.6 g b.i.d.: −0.55 mEq/L Moderate group, reduction in K^+^: - 8.4 g b.i.d.: −0.87 mEq/L - 12.6 g b.i.d.: −0.97 mEq/L - 16.8 g b.i.d.: −0.92 mEq/L Adverse events: hypomagnesemia (7.2%), mild to moderate constipation (6.3%) and hypokalemia (5.6%).
Pitt et al., 2011 [57] PEARL HF trial	105	HF pts, history hyperkalaemia resulting in discontinuation of a RAASi and/or beta-adrenergic blocking agent or eGFR < 60 mL/min	RCT	30 g o.d. RLY5016 or placebo for 4 weeks. Spironolactone, initiated at 25 mg o.d., increased to 50 mg o.d. on Day 15 if K^+^ was ≤5.1 mEq/L.	4 weeks	RLY5016 significantly lowered serum K^+^ levels: difference between groups −0.45 mEq/L (*p* < 0.001) Lower incidence HK (7.3% RLY5016 vs. 24.5% placebo, *p* = 0.015) Higher proportion pts on spironolactone 50 mg o.d.: 91% RLY5016 vs. 74% placebo, *p* = 0.019. In CKD: difference in K^+^ between groups: −0.52 mEq/L (*p* = 0.031) Incidence HK: 6.7% RLY5016 vs. 38.5% placebo (*p* = 0.041). Adverse events: mainly mild or moderate GI. Hypokalaemia in 6% of RLY5016 pts vs. 0% of placebo pts.
Pitt et al., 2018 [58] AMETHYST-DN trial subanalysis	105	Pts type 2 DM, CKD, and HK [K^+^] > 5.0–5.5 mEq/L (mild) or >5.5–<6.0 mEq/L (moderate)], with or without HF, on RAASi	RCT	Stratified by baseline serum K^+^ into mild or moderate HK groups and received 1 of 3 randomized starting doses of patiromer (4.2 g b.i.d., 8.4 b.i.d., or 12.6 g b.i.d. (mild HK0 or 8.4 g b.i.d., 12.6 g b.i.d., or 16.8 g b.i.d. (moderate HK)).	52 weeks	In HF patients, mean serum K^+^ decreased by Day 3 through Week 52. At Week 4: - Mild group: −0.64 mEq/L - Moderate group: −0.97 mEq/L Normokalemia: - Mild group: >88% - Moderate group: ≥73% The most common adverse event: hypomagnesaemia (8.6%).
Zhuo et al., 2022 [61]	3965	New-user cohort study non-dialysis adults who initiated SZC or patiromer	Retrospective observational study	Comparing SZC vs. patiromer in HHF occurrence	150 days	SZC group: 88 cases of HHF (incidence: 35.8 per 100 person-years) Patiromer group: 245 cases of HHF (incidence: 25.1 per 100 person-years). Rate HHF higher in SZC than patiromer initiators (HR: 1.22, 95% CI 0.95–1.56), but not statistically significant.
Kovesdy et al., 2019 [62]	10126	HD patients who had received patiromer, SPS, or laboratory evidence of hyperkalemia (NoKb cohort)	Retrospective observational study	527 (patiromer) 852 (SPS) 8747 (NoKb) HD patients.	141 days	Patiromer initiators had multiple prior HK (OR 2.6, 95% CI 1.8–3.7). 61% started with patiromer 8.4 g o.d. Reductions in K^+^: −0.5 mEq/L
Kovesdy et al., 2020 [63]	288	Veterans with HK (K^+^ ≥ 5.1 mEq/L)	Retrospective observational study	Patiromer initiators	6 months	K^+^ reductions post-patiromer initiation: −1.0 mEq/L- At 3–6 months: K^+^ < 5.1 mEq/L: 71% of pts K^+^ < 5.5 mEq/L: 95% of pts RAASi continued in >80–90% of patiromer-treated patients.
Piña et al., 2020 [64]	653	HF and HK	Meta-analysis RCTs	Starting doses of patiromer ranged from 8.4 to 33.6 g o.d.	4 weeks	Serum K^+^ decreased to <5.0 mEq/L within 1 week, nadir after 3 weeks in both HF and non-HF subgroups (4.59 mEq/L and 4.64 mEq/L, respectively). At 4 weeks: serum K^+^ difference from baselines: −0.79 ± 0.06 mEq/L in HF pts and −0.75 ± 0.02 mEq/L in non-HF pts. Adverse event in 31% HF pts and 37% non-HF pts: constipation (HF pts: 7%, non-HF pts: 5%) and diarrhea (HF pts: 2%, non-HF pts: 4%).

Abbreviations: b.i.d., bis in die; CKD, chronic kidney disease; g, grams; CI, confidence interval; eGFR, estimated glomerular filtration rate; HD, haemodialysis; HF, heart failure; HHF, heart failure hospitalization; HK, hyperkalemia; HR, hazard ratio; K^+^, serum concentration potassium; NoKb, no K^+^ binder; o.d., once daily; pts, patients; O.R., odds ratio; RAASi, renin–angiotensin–aldosterone system inhibitor; RCT, randomized controlled trials; SPS, sodium polystyrene sulfonate; SZC, sodium zirconium cyclosilicate.

## 5. Current Indication and Future Perspectives

The introduction of novel potassium binders have improved the overall number of pharmacological weapons in counteracting HK. The FDA and the EMA approved SZC in 2018 [32,65], while patiromer was approved by the FDA in 2015 [52] and by the EMA in 2017 [53].

Current research indicates the implementation of such compounds in clinical practice for the treatment of hyperkalemia in adults, while there is no indication for the adoption of either SZC or patiromer in the emergency setting [32,52,53,65]. SZC is recommended at a starting dose of 10 g t.i.d. for up to 48 h, and then 10 g o.d.; a reduction in dose at 5 g o.d. should be considered in relation to serum K^+^ target levels [32,65]. Indeed, patiromer is recommended at a starting dose of 8.4 g o.d., with variations in dosages to be performed in relation to serum K^+^ [52,53]. By analyzing the data in the literature, each compound was able to provide a mean reduction in serum K^+^ levels, higher than 15% as compared with the baseline (Figure 1).

The overall burden of adverse side effects related to the use of new potassium binders seems to not significantly impact on the current administration of SZC and patiromer in patients with HK, and with HF in need for implementation of RAASi therapy in particular, as compared with the old-fashioned administration of SPS. Although the risk of necrosis seem to be not so higher as compared with controls, SPS still continues to show an increased risk for gastrointestinal side effects [40]. SZC mostly induces oedema as a major side effect, while hypokalemia event rates may vary according to the dosages and the baseline serum K^+^ levels; no significant adverse gastrointestinal events have been identified in the different studies [64]. Patiromer could better promote metabolic alterations due to the impact on electrolytes, while non-serious miscellaneous (pruritus, fatigue, and headache) events might be better controlled by adopting a low dose of the drug [66]. Furthermore, it should be started 3 h after the assumption of positively charged drugs (e.g., ciprofloxacin, levothyroxine, and metformin) due to interaction in the absorption of these compounds [53].

The adoption of these drugs in acute settings is not allowed according to the current indications from the summary of product characteristics (SmPC) of both of the drugs [32,52,53,65]. Patiromer was used at a dose of 25.2 g in adult patients with end-stage renal disease and a serum K^+^ level ≥ 6.0 mEq/L in adjunct with standard-of-care (vs. standard-of-care alone): K^+^ levels were slightly reduced at 2 h post-administration, but the differences were lost at 6-hour follow-up [67]. Indeed, Di Palo et al. [68] observed that a single dose of oral patiromer (8.4 g, 16.8 g, or 25.2 g in agreement with baseline serum K^+^ levels) induced a significant reduction of about 0.5 mEq/L in serum K^+^ levels within 6 h, when acutely administered in patients with HK. Nevertheless, no comparisons with standard therapies, no life-threatening conditions, and a 6-hour delay in answering to therapy represent the main limitations of research in this setting. Finally, patiromer was less able than SPS to reduce HK in an acute setting as demonstrated by Nguyen et al. [69], thus, more studies are needed in the context of an urgency response. According to SZC, the ENERGIZE trial [70] randomized patients with serum K^+^ levels > 5.8 mEq/L to insulin and glucose and SZC 10 g or placebo, up to three times during a 10-hour period. The results demonstrated a significant reduction in serum K^+^ levels within 2 h, with comparable adverse effects between the two arms. Nonetheless, as this is only a phase II trial, further protocols are needed in order to evaluate the application of SZC in acute settings.

Indeed, new potassium binders might be extremely attractive in the chronic management of patients with HF who need implementation of RAASi therapies. Long-term results seemed to positively promote the implementation of these compounds in clinical practice due to their low rate of side effects and absence of gastrointestinal absorption. Multiple dosages can modulate therapies in relation to the serum K^+^ levels, thus, promoting tight control of patients with HF and amelioration in the management of this worrisome disease. Figure 2 shows a proposed flow chart to be considered for the management of patients with HF and HK. The inclusion of novel potassium binders could provide the opportunity for implementing RAASi therapy in patients who cannot attain target doses due to increased values in serum K^+^ levels.

The impact on outcomes, adverse reactions due to prolonged therapy with these compounds (if used more than one year), and effective place on therapy will surely be the next topics within the HF field of research.

## 6. Conclusions

Hyperkalemia can impact the management of patients with HF by promoting the discontinuation of therapies, thus, negatively increasing the risk for mortality. The adoption of novel potassium binders, namely SZC and patiromer, could possibly overcome morbidities and mortality related to HK in HF patients. Further studies are needed in order to promote the effective impact of these novel pharmacological compounds to modify the natural history of HF.

## Figures and Tables

**Figure 1 biomedicines-10-01721-f001:**
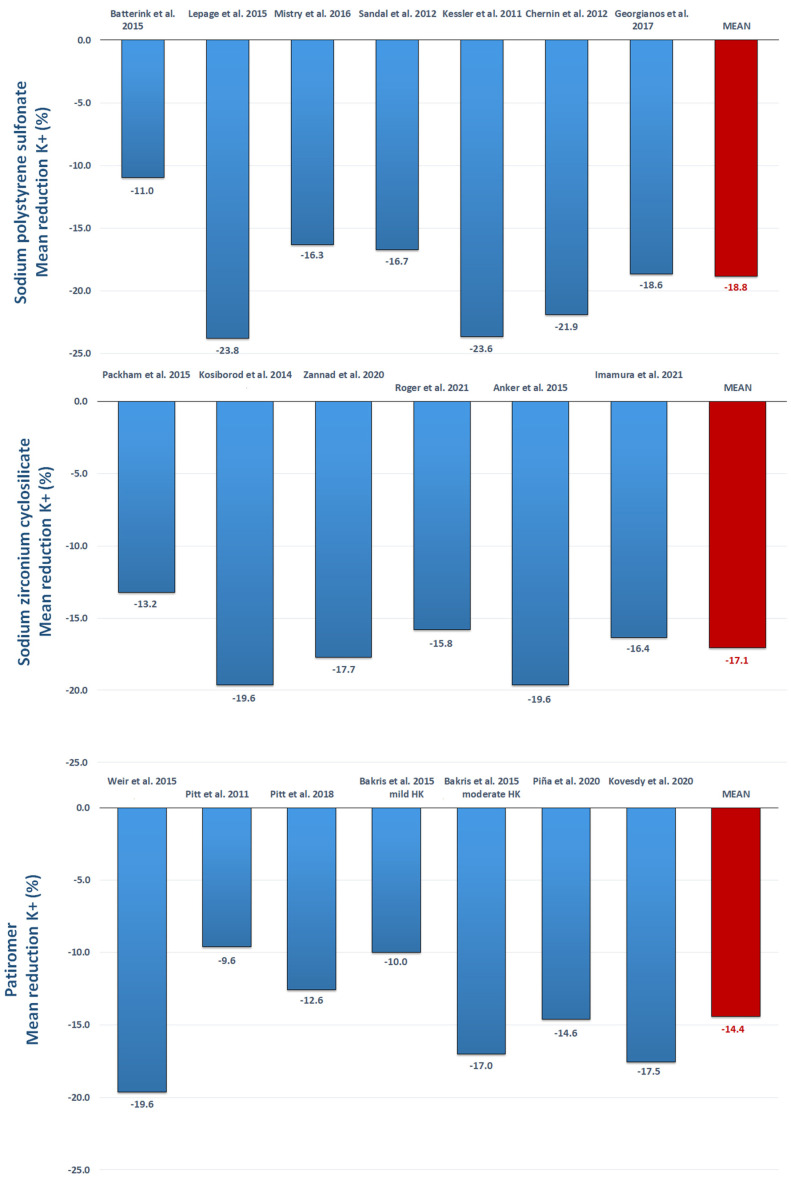
Schematic representation of mean percentage reduction in serum concentration of potassium with the three potassium binders (sodium polystyrene sulfonate, sodium zirconium cyclosilicate, and patiromersorbitex calcium): A reappraisal from literature [33,34,35,36,37,38,39,42,43,44,46,48,49,55,56,57,58,60,63].

**Figure 2 biomedicines-10-01721-f002:**
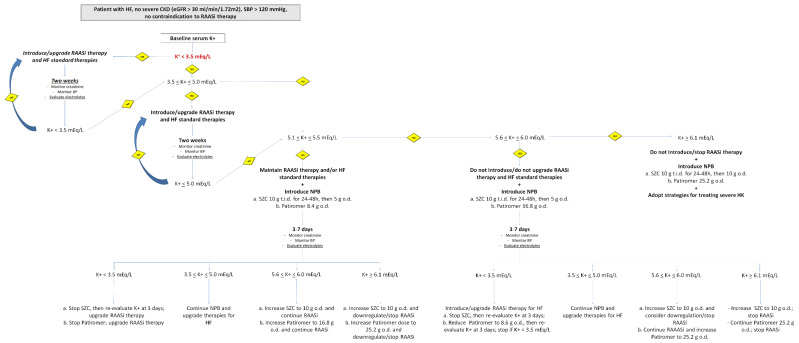
Proposed flow chart for the implementation of renin–angiotensin–aldosterone system inhibitors (RAASi) in patients with heart failure (HF), with no severe chronic kidney diseases (estimated glomerular filtration rate (eGFR) > 30 mL/min/m^2^) and hyperkalemia (serum K^+^ > 5.1 mEq/L).

**Table 1 biomedicines-10-01721-t001:** Main characteristics of potassium binders: Sodium polystyrene sulfonate, sodium zirconium cyclosilicate, and patiromer sorbitex calcium.

Characteristics	Sodium Polystyrene Sulfonate	Sodium Zirconium Cyclosilicate	Patiromer Sorbitex Calcium
**Chemical formula**	[C_8_H_8_SO_3_^−^]_n_	(2Na·H_2_O·3H_4_SiO_4_·H_4_ZrO_6_)_n_	[(C_3_H_3_FO_2_)182·(C_10_H_10_)8·(C_8_H_14_)10]n [Ca91(C_3_H_2_FO_2_)182·(C_10_H_10_)8·(C_8_H_14_)10]n (calcium salt)
**Chemical structure**	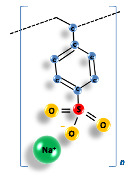	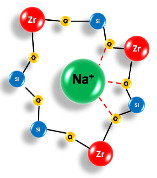	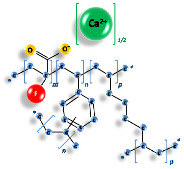
**Molecular weight**	184.21 U	371.5 U	901.10
**Administration**	Oral or rectal administration	Oral administration	Oral administration
**Dose**	Oral: 15 g to 60 g 1 to 4 times daily. Rectal: 30 to 50 g every 6 h.	Starting: 10 g t.i.d. Maintenance: 5 g o.d. (eventually increase to 10 g o.d.) *In hemodialysis:* 5 g o.d. (till 15 g o.d.) in non-dialysis days	Starting dose is 8.4 g patiromer o.d. Daily dose may be increased to 16.8 g or maximum 25.2 g o.d.
**Absorption**	None	None	None
**Excretion**	Feces	Feces	Feces
**Onset of Action**	Within 2–24 h till 4 to 6 h Exchange capacity: ~33% or 1 mEq of K^+^ per 1 g of resin. Variation for competition to other cations (Na^+^, Ca^2+^, Mg^2+^)	Within 1–6 h, normokalemia in 24–48 h	Within 4–7 h, duration about 24 h
**Pharmacodynamics**	Cation exchange resin, Na^+^ ions partially released from polystyrene and replaced by K^+^	Non-absorbed, non-polymer inorganic powder with a micropore structure that high selectively captures K^+^ in exchange for H^+^ and Na^+^ in the GI tract.	Non-absorbed, cation exchange polymer that contains a calcium-sorbitol complex. It binds K^+^ in the lumen of the GI tract.
**Side effects**	↑ [Na^+^]; ↓ [Ca^2+]^; ↓ [K^+^]; ↓ [Mg^2+^] GI: Anorexia, constipation, diarrhea, fecal impaction, nausea, vomiting <1%, postmarketing, and/or case reports: Bezoar formation, GI hemorrhage, GI ulcer, intestinal necrosis, intestinal perforation, ischemic colitis	↓ K^+^ Edema GI: anorexia, constipation, diarrhea, nausea, vomiting	↓ Mg^2+^ GI disorders: constipation, diarrhoea, abdominal pain, flatulence, nausea, vomiting

Abbreviations: [Ca^2+^], serum concentration calcium; GI, gastrointestinal; h, hours; [Na^+^], serum concentration sodium; [K^+^], serum concentration potassium; [Mg^2+^], serum concentration magnesium; o.d., once daily; t.i.d., ter in die. ↑: increase; ↓: decrease.

**Table 3 biomedicines-10-01721-t003:** Main characteristics of studies on sodium zirconium cyclosilicate.

Study	N. of pts	Type of pts	Design	Approach	Follow-Up	Results
Packham et al., 2015 [42]	754	Patients with K^+^ 5.0–6.5 mEq/L	RCT	Randomly assigned to 1.25 g, 2.5 g, 5 g, or 10 g of SZC or placebo t.i.d. for the initial 48 h (initial phase). Those in the SZC group who reached K^+^ 3.5–4.9 mEq/L at 48 h randomly assigned (1:1) to original SZC dose or placebo o.d. on days 3 to 14 (maintenance).	14 days	At 48 h K^+^ decreased: - Group 1.25 g t.i.d.: 5.3 mEq/L to 5.1 mEq/L - Group 2.5 g t.i.d.: 5.3 mEq/L to 4.9 mEq/L - Group 5 g t.i.d.: 5.3 mEq/L to 4.8 mEq/L - Group 10 g t.i.d.: 5.3 mEq/L to 4.6 mEq/L Maintenance phase: - 5 g and 10 g maintained K^+^ to 4.7 mEq/L and 4.5 mEq/L, respectively Adverse events similar in SZC and placebo group (12.9% vs. 10.8%, initial phase; 25.1% vs. 24.5%, maintenance phase). Diarrhea was the most common.
Kosiborod et al., 2014 [43] HARMONIZE trial	258	Outpatients with K^+^ ≥ 5.1 mEq/L	rct	10 g szc t.i.d. in the initial 48-h, Those achieving k^+^ 3.5–5.0 mEq/L randomized to szc 5 g, 10 g, or 15 g, or placebo o.d. for 28 days	28 days	At 48h K^+^ decreased from 5.6 mEq/L to 4.5 mEq/L Normokalemia in 84% in 24 h; 98% in 48 h. Days 8–29 (vs. placebo): - Group 5 g t.i.d.: 4.8 mEq/L vs. 5.1 mEq/L - Group 10 g t.i.d.: 4.5 mEq/L vs. 5.1 mEq/L - Group 15 g t.i.d.: 4.4 mEq/L vs. 5.1 mEq/L N. of pts with K^+^ < 5.1 mEq/L at Days 8–29: - Group 5 g t.i.d.: 80% - Group 10 g t.i.d.: 90% - Group 15 g t.i.d.: 94% Adverse events comparable between SZC and placebo; edema more common in the 15 g group; hypokalemia in 10% of the 10 g group and 11% of the 15 g group, none in the 5 g or placebo groups.
Zannad et al., 2020 [44] HARMONIZE GLOBAL	262	Outpatients with K^+^ ≥ 5.1 mEq/L	RCT	10 g SZC t.i.d. in the initial 48 h, Those achieving K^+^ 3.5–5.0 mEq/L randomized to SZC 5 g, 10 g, or placebo o.d. for 28 days	28 days	92.9% reached normokalemia after 48 h; mean reduction in K^+^: −1.28 mEq/L vs. baseline (*p* < 0.001). Days 8–29, mean reduction K^+^: - Group 5 g t.i.d.: 9.6% - Group 10 g t.i.d.: 17.7% N. of pts with K^+^ < 5.1 mEq/L at Days 8–29: - Group 5 g t.i.d.: 58.6% - Group 10 g t.i.d.: 77.3% Most common adverse events with SZC: mild or moderate constipation and oedema.
Roger et al., 2019 [45] HARMONIZE extension	123	HARMONIZE trial pts with K^+^ 3.5–6.2 mEq/L	RCT	SZC 5–10 g o.d. for ≤337 days	337 days	K^+^ ≤ 5.1 mEq/L in 88.3% of pts after 337 days K^+^ ≤ 5.5 mEq/L in 100% of pts after 337 days
Roger et al., 2021 [46]	751	Outpatients with K^+^ ≥ 5.1 mEq/L and Stages 4 and 5 CKD versus those with Stages 1–3 CKD.	RCT	SZC 10 g t.i.d. for 24–72 h until K^+^ 3.5–5.0 mmol/L then SZC 5 g o.d. for ≤12 months Patients stratified by eGFR < 30 or ≥30 mL/min/1.73 m^2^	12 months	Percentage of pts with normokalemia: - 82% of pts in both eGFR within 24 h - 100% of patients with eGFR < 30 mL/min/1.73 m^2^ within 72 h - 95% of patients with eGFR ≥ 30 mL/min/1.73 m^2^ within 72 h - 82% of patients with eGFR < 30 mL/min/1.73 m^2^ within 365 days - 90% of patients with eGFR ≥ 30 mL/min/1.73 m^2^ within 365 days
Fishbane et al., 2019 [47] DIALIZE trial	196	ESRD in 3-times weekly hemodialysis and predialysis hyperkalemia	RCT	Randomized to placebo or SZC 5 g o.d. (titrated till 15 g in relation to serum K^+^ level) on non-dialysis days.	4 weeks	41.2% reached normokalemia. Serious adverse events in 7% pts treated with SZC Few episodes of hypokalemia.
Anker et al., 2015 [48] HARMONIZE substudy	94	HF pts from HARMONIZE, with serum K^+^ ≥ 5.1 mEq/L, and including those receiving RAASi.	RCT	Open-label SZC for 48 h. Those who achieved K^+^ 3.5–5.0 mEq/L randomized to SZC 5, 10, or 15 g or placebo o.d. for 28 days.	28 days	Despite RAASi doses being kept constant, serum K^+^ levels were: - Group 5 g o.d.: 4.7 mEq/L - Group 10 g o.d.: 4.5 mEq/L - Group 15 g o.d.: 4.4 mEq/L Percentage pts with normokalemia: - Group 5 g o.d.: 83% - Group 10 g o.d.: 89% - Group 15 g o.d.: 92%
Imamura et al., 2021 [49]	24	HF pts with LVEF < 50% and hyperkalemia	Retrospective observational study	SZC 5–15 g o.d.	3 months	↓ serum K^+^ ↑ RAASi dose No adverse events

Abbreviations: CKD, chronic kidney disease; g, grams; eGFR, estimated glomerular filtration rate; ESRD, end-stage renal disease; h, hours; HF, heart failure; K^+^, serum concentration potassium; LVEF, left ventricle ejection fraction; o.d., once daily; pts, patients; RAASi, renin–angiotensin–aldosterone system inhibitor; RCT, randomized controlled trials; SZC, sodium zirconium cyclosilicate; t.i.d., ter in die. ↑: increase; ↓: decrease.

## Data Availability

Not applicable.

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
