# Peer review of "Optimizing Therapies in Heart Failure: The Role of Potassium Binders"

_biomedicines, 2022, doi:10.3390/biomedicines10071721_

Round 1

Reviewer 1 Report

In this manuscript, the authors reviewed about new Potassium binders for Heart failure patients with less severe CKD. The efforts are praiseworthy as the manuscript could be used as a learning material by many healthcare professionals. Here are some minor concerns:

1. Line 123,124 needs English check. 

2. Table 1 is excellent. However, the parts need to properly spaced as the texts of "administration", "Dose", "absorption" are clustered heavily.  

3.Line 146, 150 needs English check. 

4. Table 2 to table 4 the fonts are smaller than table 1. Table 2 to 4 could have similar larger font as table 1. 

5. Line 187 to 190 need some reference. 

6. In table 3. in Kosiborod et al part the on the right side Group 10 g t.i.d showed 5.1.6mEq/L. The reference says 5.1mEq/L. Needs to be corrected. 

7. In Figure 01, the lower panel showed .9.6 in Pitt et al 2011 needs to be corrected. How the "mean" has been calculated and what concluded from this mean, that is needed to be mentioned. 

8. Line 324 says "KH". This KH was also used in Table 4 in Kovesdy et al part. What is this KH? An abbreviation and explanation is needed. Otherwise needs to be corrected. 

9. Figure 02 needs more attention. The fonts are very small, barely visible. All the place it is written "Monitora BP". Again, the texts are heavily clustered and there is ampule space in the bottom. 

Author Response

Reviewer #1

We thank this Reviewer for the constructive comments and suggestions. Furthermore, we would like to really thank him/her for his/her appreciation about our research. This is our point to point reply.

In this manuscript, the authors reviewed about new Potassium binders for Heart failure patients with less severe CKD. The efforts are praiseworthy as the manuscript could be used as a learning material by many healthcare professionals. Here are some minor concerns:

Thank you for the appreciation of our work.

  1. Line 123,124 needs English check.

Done

  1. Table 1 is excellent. However, the parts need to properly spaced as the texts of "administration", "Dose", "absorption" are clustered heavily.

Thank you very much for your suggestion. We revised the table 1 in order to improve its clarity.

3.Line 146, 150 needs English check.

Done.

  1. Table 2 to table 4 the fonts are smaller than table 1. Table 2 to 4 could have similar larger font as table 1.

Thank you for the suggestion. We adopted the same large font for all the tables.

  1. Line 187 to 190 need some reference.

We included reference 41 to address the content of the sentences.

  1. In table 3. in Kosiborod et al part the on the right side Group 10 g t.i.d showed 5.1.6mEq/L. The reference says 5.1mEq/L. Needs to be corrected.

Sorry for the mistake and thank you for identifying the typo. We amend it.

  1. In Figure 01, the lower panel showed .9.6 in Pitt et al 2011 needs to be corrected. How the "mean" has been calculated and what concluded from this mean, that is needed to be mentioned.

Thank you once again for this comment. We amend the wrong by substituting .9.6 with “-9.6”. “mean” stands for the mean percentage reduction in serum concentrations of K+ from baseline to the end of the administration period. The data are derived from the corresponding literature and were computed in order to better delineate the average reduction in K+ which was performed by each compound.

  1. Line 324 says "KH". This KH was also used in Table 4 in Kovesdy et al part. What is this KH? An abbreviation and explanation is needed. Otherwise needs to be corrected.

Once again sorry for the mistake. KH stands for HK, i.e. hyperkalemia. We amended it in the text. Thanks for the remark.

  1. Figure 02 needs more attention. The fonts are very small, barely visible. All the place it is written "Monitora BP". Again, the texts are heavily clustered and there is ampule space in the bottom.

Thank you for the comment. We amended the typos in the figure. Indeed, the small dimension of the figure is linked to the need for entering it into the word/pdf file. We uploaded the figures in TIF in order to improve the readability of the text. In our PPT format, the font dimension is 28 which is really visible but the translation of the figure in the word file reduces the identification of the words. I hope that the Editorial Office could give me a hand for ameliorating the visualization of the figures.

Reviewer 2 Report

I congratulate the authors for approaching this topic of major clinical interest for practitioners. The review has a coeherent thesis that is well described and structured, based on up-dated references. Clinical data are also synthesized in appropriate tables. It is a balanced manner of reporting and discussing the data, highlighting adverse side-effects related to the use of new potassium-binders.

An obvious strength is that the authors computed literature data and provided an objective representation of mean percentage reduction in serum concentration of potassium with the 3 different potassium-binders.

I congratulate the authors for proposing a practical flow-chart for the management of hiperkaliemic patients with HF.

I recommend the authors to include the methodology for selecting the reported studies. Please provide how this review article will benefit the clinicians as an update.

Author Response

Reviewer #2

We thank this Reviewer for her/his useful suggestions. We sincerely appreciate his/her comments on our work. This is our point-to-point reply:

I congratulate the authors for approaching this topic of major clinical interest for practitioners. The review has a coeherent thesis that is well described and structured, based on up-dated references. Clinical data are also synthesized in appropriate tables. It is a balanced manner of reporting and discussing the data, highlighting adverse side-effects related to the use of new potassium-binders. An obvious strength is that the authors computed literature data and provided an objective representation of mean percentage reduction in serum concentration of potassium with the 3 different potassium-binders. I congratulate the authors for proposing a practical flow-chart for the management of hiperkaliemic patients with HF.

Thank you very much for the approval of this manuscript and its contents. We really appreciate it.

  1. I recommend the authors to include the methodology for selecting the reported studies. Please provide how this review article will benefit the clinicians as an update.

We would like to really thank the reviewer for his/her comments. We included a dedicated paragraph for describing the methodology for searching and selecting the studies. We also include few lines in order to better outline the possible impact of this review in daily clinical practice.